# Screening Evaluation of Antiproliferative, Antimicrobial and Antioxidant Activity of Lichen Extracts and Secondary Metabolites In Vitro

**DOI:** 10.3390/plants12030611

**Published:** 2023-01-30

**Authors:** Martin Kello, Michal Goga, Klaudia Kotorova, Dominika Sebova, Richard Frenak, Ludmila Tkacikova, Jan Mojzis

**Affiliations:** 1Department of Pharmacology, Faculty of Medicine, Pavol Jozef Šafárik University, 040 01 Košice, Slovakia; 2Department of Botany, Institute of Biology and Ecology, Faculty of Science, Pavol Jozef Šafárik University, 041 67 Košice, Slovakia; 3Department of Microbiology and Immunology, University of Veterinary Medicine and Pharmacy, 041 81 Košice, Slovakia

**Keywords:** lichens, secondary metabolites, cytotoxicity, 2D, 3D spheroids, antibacterial, antioxidant

## Abstract

Lichen metabolites represent a wide range of substances with a variety of biological effects. The present study was designed to analyze the potential antiproliferative, antimicrobial and antioxidative effects of several extracts from lichens (*Pseudevernia furfuracea*, *Lobaria pulmonaria*, *Cetraria islandica*, *Evernia prunastri*, *Stereocaulon tomentosum*, *Xanthoria elegans* and *Umbilicaria hirsuta*) and their secondary metabolites (atranorin, physodic acid, evernic acid and gyrophoric acid). The crude extract, as well as the isolated metabolites, showed potent antiproliferative, cytotoxic activity on a broad range of cancer cell lines in 2D (monolayer) and 3D (spheroid) models. Furthermore, antioxidant (2,2-diphenyl-1-picryl-hydrazylhydrate (DPPH) and in vitro antimicrobial activities were assessed. Data showed that the lichen extracts, as well as the compounds present, possessed biological potential in the studied assays. It was also observed that the extracts were more efficient and their major compounds showed strong effects as antiproliferative, antioxidant and antibacterial agents. Moreover, we demonstrated the 2D and 3D models’ importance to drug discovery for further in vivo studies. Despite the fact that lichen compounds have been neglected by the scientific community for long periods, nowadays they are objects of investigation based on their promising effects.

## 1. Introduction

Despite significant scientific progress, cancer is one of the most common causes of death in the world. Patients with cancer suffer from serious health problems, which often end in death, even though current health care for oncology patients reduces worldwide mortality and increases survival every year. In 2019, statistics showed a 32% overall drop in death risk, which is connected with reductions in smoking (lung cancer), adjuvant chemotherapies (colon and breast cancer), combination therapies (cancers in general), advances in early detection, surgical techniques and targeted therapies [1]. However, it is assumed that the number of new cancer cases will rise by approximately 50% worldwide over the next two decades. In 2020, over 19 million newly diagnosed patients appeared worldwide, of which 10 million died [2]. To fight cancer, several approaches have been developed including surgery, chemotherapy, radiation, immunotherapy and targeted therapy. Currently, chemotherapy represents the basic approach in the treatment of cancer, in spite of the fact that synthetic chemotherapy drugs often cause serious adverse effects. Therefore, new treatments and approaches in cancer research are challenging the contemporary research community.

An important role in tumor treatment could be played by natural products (NP). Plants are still the basic source of NP for the treatment of many diseases, including cancer. The antitumor properties of natural products are undergoing intensive research because of their beneficial effects on tumor inhibition, reduced radiotherapy and chemotherapy side effects, fewer adverse reactions and less drug resistance, and prolonged survival. In the drug industry, lichens are the sources of more than 1000 anticancer active compounds with a broad spectrum of biological activities including anti-proliferative, pro-apoptotic, anti-migratory and anti-metastatic, as reviewed in [3,4,5]. Moreover, antioxidant and antibacterial properties of some lichen extracts and metabolites have already been reported [6,7,8,9]. Lichens produce unique secondary metabolites (depsides, depsidones, pulvinic acid derivatives, dibenzofurans) and pigments (anthraquinones, napthoquinones and xanthones) [10]. In this paper, the extracts of lichen species *Pseudevernia furfuracea*, *Lobaria pulmonaria*, *Cetraria islandica*, *Evernia prunastri*, *Stereocaulon tomentosum*, *Xanthoria elegans*, *Umbilicaria*, *hirsuta* and some of their secondary metabolites, such as gyrophoric acid, evernic acid, atranorin (depsids) and physodic acid (depsidones), were studied for their antiproliferative potential. The standard method for potential anticancer drug discovery has been High-Throughput Screening (HTS), which enables fast, effective testing of a large number of biological modulators and effectors against selected and specific targets [11]. For drug discovery, cell-based assays play an important role and give information about the acute cytotoxicity profile and the ability of the chemical compounds to penetrate the cell membrane. Two-dimensional (2D) cell cultures, despite the above mentioned advantages, also have some limitations such as the fact that they cannot imitate the natural structures of tissues or tumors; the natural tumor microenvironment and cell–extracellular environment interactions are not fully presented [12,13]. Therefore, an alternative model, better able to mimic a natural tumor mass, was established. Three-dimensional (3D, spheroids) cell cultures represent simple in vitro interlinks between in vitro 2D and in vivo tumor models. In spheroid models, cells grow with a 3D microarchitecture in various layers, mimicking tissue organization, with better cell-to-cell trafficking and signaling networks [14]. In drug discovery screening, 3D models with the same cell density as tissue show a comparable drug response to solid tumors [15]. Therefore, comparison of 2D and 3D tumor screening of lichen extracts and secondary metabolites forms the main part of this paper. Because lack of information exists about 3D screening of lichen substances, in this manuscript we focus on comparison of 2D and 3D cytotoxic assays. Moreover, antibacterial properties of crude extracts and their most abundant metabolites were tested with the addition of the antioxidant properties of these extracts. The choice of extracts and corresponding secondary metabolites were based mostly on published information and the absence of experimental data.

## 2. Results and Discussion

It is a fact that natural substances have been of great interest over several decades as a source of potential multitarget cancer chemotherapeutic agents with varied mechanisms of action. Until 2019, almost 300 small-molecule substances approved in Western medicine as antitumor drugs were originally either natural products or synthetic derivatives based on natural products or were inspired by a natural substance as a template for drug engineering [16]. Despite these facts, new anticancer drug identification remains important and crucial for cancer research and therapy. A very potent source of natural substances, with over 1000 unique secondary metabolites of various chemical structures, are lichens. Recently, their pro-apoptotic, antiproliferative, antioxidant and anticancer properties have been intensively studied [3,5,17,18]. Therefore, in our work, we pay attention to the possible use of extracts from the lichen species *Pseudevernia furfuracea*, *Lobaria pulmonaria*, *Cetraria islandica*, *Evernia prunastri*, *Stereocaulon tomentosum*, *Xanthoria elegans* and *Umbilicaria hirsuta*. Secondary metabolites of these extracts were analysed and identified previously by TLC, HPLC and LC/MS methods, as summarized in Table 1. Some of their most frequently occurring (apart from the well-studied usnic acid) or interesting secondary metabolites, such as gyrophoric acid, physodic acid, evernic acid and atranorin, were analysed as antitumor and antibacterial agents and potential antioxidants. We focused on complex evaluation of potential effects of these extracts and secondary metabolites, as scientific knowledge is lacking in the given area.

### 2.1. Antiproliferative Activity

The metabolic resazurin assay was used to evaluate cytotoxic potential of the whole spectrum of tested extracts and secondary metabolites in two different cultivation conditions: 2D cell culture monolayers or suspension culture and 3D spheroids. There is clear evidence that 3D spheroids mimic the natural structures of tissues or tumors with their proliferative surface and necrotic core and the different diffusion and distribution of substances or nutrients, cell-to-cell connections and signaling [30]. Therefore, compared to 2D cultures, spheroids must be taken into consideration as interlinks between in vitro and in vivo tissue conditions, which contribute to basic anticancer drug screening and closer to in vivo concentrations’ determination. In the research area of lichen extracts and secondary metabolites screening, any data from 3D spheroids is almost exclusively missing, except in a recent published paper by Majchrzak-Celińska and colleagues which used glioblastoma cells’ (T98G) spheroids [31].

The results from screening tests clearly showed different levels of effectiveness of the tested substances across the entire spectrum of tested cell lines (Table 1 and Table 2) in both 2D and 3D spheroids’ conditions. In the 2D system (Table 1), analyses revealed that the best source of potential cytotoxic substances would be extracts from *Lobaria pulmonaria* (37.2–225.3 µg/mL), *Evernia prunastri* (43.5–128.0 µg/mL), *Stereocaulon tomentosum* (44.0–146.5 µg/mL) and *Pseudevernia furfuracea* (31.9–48.6 µg/mL) and secondary metabolite physodic acid, with the lowest relative concentration range. These lichens along with physodic acid showed effective inhibition of metabolic activity on all tested cancer cell lines, mostly with IC_50_ in the range 31.9–225.3 µg/mL (extracts) or 37.7–162.1 µM (physodic acid). Moreover, comparison to fibroblast or normal epithelial cells showed partial selectivity specifically based on cell type and extract/metabolite origin. As shown in Table 2, selectivity cannot be generalized across the whole spectrum and must be examined individually. As reference chemotherapy, cisplatin (6.3–35.4 µM) was used, where comparison with extracts/metabolites showed close or higher IC_50_. From the view of cancer cell line tissue origin and the number of extracts/metabolites with IC_50_ under 100 µg/mL or 100 µM, the most sensitive cell lines were A2780 (7 substances) > HeLa = Jurkat (6) > HCT116 = MDA-MB-231 (5) > A549 = BLM (4) ˃ Caco-2 (3) ˃ MCF-7 (2). In addition, DMSO vehicle test (Figure 1A) showed significant metabolic inhibition, mostly in concentrations above 0.2% *v*/*v*. We provided also calculated IC_50_ values for DMSO (Table 2) showing hypothetical values mimicking extract or secondary metabolite concentrations when the solvent reached 50% cytotoxicity. Above 0.2% *v*/*v* of DMSO, which is approximately equivalent to 100 µM or 100 µg/mL, the solvent toxicity must be taken into consideration in estimating benefits or risks.

In 3D spheroid models (Table 3), the best inhibition effects were shown by the same extracts as in the 2D model, *Lobaria pulmonaria* (8.3–150.8 µg/mL), *Evernia prunastri* (46.7–551.2 µg/mL), *Stereocaulon tomentosum* (44.5–552.0 µg/mL) and *Pseudevernia furfuracea* (19.5–129.0 µg/mL), with the addition of *Umbilicaria hirsuta* extract (9.3–143.3 µg/mL) and secondary metabolite gyrophoric (17.9–148.6 µM) acid. Physodic acid in the 3D spheroid model showed significantly higher IC_50_ values compared to 2D. We cannot generally conclude from 3D spheroids’ results that three-dimensional cultivation and altered physiology of spheroids led to lower susceptibility to tested extracts and less effectiveness, with higher IC_50_ values. The best example from our results is the ovarian cancer cell line A2780 where in all cases 3D IC_50_ values increased several times compared to the 2D model. Furthermore, cisplatin IC_50_ (6.7–183.8 µM) was most raised compared to 2D model. From the view of cancer cell line tissue origin and the number of extracts/metabolites with 3D IC_50_ under 100 µg/mL or 100 µM, the most sensitive cell lines were MDA-MB-231 (seven substances) ˃ MCF-7 = HeLa (6) ˃ HCT116 = A549 (5) ˃ Jurkat (3) ˃ Caco-2 = BLM (2) ˃ A2780 (1). Paradoxically, the DMSO vehicle 3D test (Figure 1B) showed significant metabolic inhibition from lower doses of 0.02% *v*/*v*. Compared to the 2D model, the hypothetical values mimicking extract or secondary metabolite concentrations when the solvent reached 50% cytotoxicity are also lower. We suggest that spheroids’ physiology strongly influences the cytotoxicity of solvent or lichen extracts/metabolites diluted in it. The main differences between the 2D and 3D models is that after 4 days of spheroid formation based on 10,000 cells, a significant fraction of the cells inside the spheroids undergo hypoxia and can naturally suffer from nutrient/oxygen deficiencies. These cells therefore probably started to die naturally, which was manifested in the fact that the 3D model compared to the 2D model showed increased cytotoxicity after DMSO or lichen extract/metabolites treatment. Increased sensitivity of cells to solvent must be taken into consideration. Alternatively, less toxic or non-toxic solvent must be used in preparation if possible.

In general, it is known that spheroids should be more resistant to chemotherapy compared to cells cultured as 2D monolayers [32]. More physiological cell–cell contacts, localization of cells in space, mass transport and mechanical properties critically influence the availability, transport and finally the cytotoxic effects of the tested substances [33]. We suggest, that the observed different 2D and 3D cytotoxic effects of the same tested extracts/metabolites depend on how the substances reached the zones of proliferating cells on the outside and the quiescent cells or necrotic core on the inside, due to nutrient and oxygen transport limitations. Penetration of substances into a spheroid is the rate-limiting step for drug delivery. Similar to in vivo tissue, some drugs may penetrate spheroids cells at non-negligible quantities only to a depth of a few cells, which is often dependent on treatment exposure time [34]. As we suggest from our 2D and 3D screening comparison of lichen extracts or secondary metabolites’ cytotoxicity in various cancer cell models, final inhibition/cytotoxic effects of the tested substances probably depended on how easily they penetrate spheroids and on their molecular structure. We also suggest that the tested substances affected the surface proliferated mass of spheroid cells in the first place, in addition to already starving, dying, dormant or senescing cells in the core. 2D adherent or suspension cell models did not suffer from such attributes. Therefore, for future in vivo experiments, such spheroid testing should be important, even though spheroids are not exactly tissues with stromal cell support, blood supply, hormonal influence, hypoxia, etc.

It is known that some chemotherapy drugs such as doxorubicin, bleomycin, 5-fluorouracil, carmustine, cisplatin, chlorambucil and mitomycin tested on the same spheroid model V79 (lung fibroblast) easily penetrate through spheroids (except for doxorubicin with poor penetration [35]). Moreover, it was proven that lichen-derived phenolic compounds (anthraquinones, dibenzofuranes, depsides, depsidones and xanthones) could interfere with several apoptotic and cell survival pathways and exert cytotoxic effects against cancer cells [29,36] or modulate cell–cell interaction in the tumor microenvironment [20] or angiogenesis [37].

### 2.2. Antibacterial Activity

All seven extracts were tested against gram positive (*Staphyloccocus aureus*) and gram negative bacteria (*Escherichia coli*). Extracts in concentration of 1mg/mL were not effective against *E. coli* in the RIZD (relative inhibition zone diameter). The results correlate with a previous study in which the two present extracts (*L. pulmonaria* and *P. furfuracea*) were tested [38]. Interestingly, against gram positive bacteria *S. aureus*, extracts from lichens *L. pulmonaria*, *C. islandica*, *E. prunastri*, *S. tomentosum* and *P. furfuracea* (Table 4) were effective. The highest inhibition zone was observed in extract of lichen *L. pulmonaria* and the lowest in extract of *E. prunastri*. From the chosen secondary metabolites, was physodic acid was most effective. The concentration of each pure compound was 0.57 mM. These results are in accordance with several published data proving the antibacterial properties of some lichens [7,9,39]. In addition, some mechanisms of the antibacterial activity of depsidones (such as the tested physodic acid) were described, suggesting a connection between their activities and RecA inhibition, which potentiates bactericidal activity [40], or the influence of β-hydroxyacyl-acyl carrier protein FabZ of the bacterial system for FAS [41]. Maciąg-Dorszyńska et al. [42] also described the general mechanism of antibacterial action of usnic acid against *Bacillus subtilis,* and S. aureus assumes inhibition of synthesis of RNA and impairment of DNA replication. It is also known that phenolic secondary metabolites of lichen with the presence of phenol hydroxyl groups are the main factors resulting in antibacterial activity [43,44] based on inhibition of extracellular microbial enzymes, oxidative phosphorylation and deprivation of the substrate necessary for microbial growth [45]. We can also hypothesize that hydroxyl groups can participate in oxidative stress and alteration of antioxidant-defense mechanisms in the cells and thus on the antibacterial/cytotoxic properties of tested lichen substances.

### 2.3. Antioxidant Activity

The DPPH radical-scavenging activity method was used to evaluate the antioxidant activities of acetone extracts of seven lichen species. Table 5 shows the DPPH scavenging activities of *E. prunastri*, *S. tomentosum*, *L. pulmonaria*, *C. islandica*, *U. hirsuta*, *X. elegans* and *P. furfuracea* in comparison to ascorbic acid. Calculated IC_50_ values ranged from 532.34 to 24.75 µg/mL. The results of the antioxidant activity of DPPH were in accordance with previous research, where in most cases an average value for free radical scavenging was achieved; however, with the extract of lichen *P. pulmonaria*, relatively good scavenging capacity was observed. It is already known from previous research that this lichen contains antioxidant active substances [21]. In the extract of the lichen *X. elegans* we observed discoloration at certain concentrations, but the resulting color of the reaction mixture (deep red) apparently absorbed the measured wavelength and the data were not applicable for expressing the inhibition of the free radicals. Extract of *C. islandica* showed the lowest antioxidant activity, which was probably caused by solvent. *C. islandica* showing better antioxidant properties by water or methanolic extraction [46]. DPPH of pure compounds (physodic acid, atranorin, evernic acid and gyrophoric acid) were tested in a previously published paper from our department [47]. None of the tested compounds showed notable free radical scavenging activity in DPPH assay compared to ascorbic acid. It is not surprising that lichen extract possesses antioxidant properties, because of its chemical composition. It is known that antioxidant properties in extracts come from the presence of phenolic compounds and/or polysaccharides with reactive (hydroxyl) functional groups [48]. The balance between antioxidant and pro-oxidant properties of lichen extracts or secondary metabolites is important for cancer treatment and for future perspectives. ROS-induced cell death followed by DNA damage mediated by lichen extracts or secondary metabolites represent an important mechanism of their action [49,50,51].

## 3. Materials and Methods

### 3.1. Collection of Lichens

Lichen *Evernia prunastri* was collected from branches of *Prunus spinosa* at Zemplínske vrchy, Vlčia hora (Cejkov, Slovakia), *Umbilicaria hirsuta* was collected from extrusive igneous volcanic rocks Sninský kameň in Vihorlat Mountains, *Pseudevernia furfuracea* was collected from spruce trees (*Picea abies*) near the village of Špania dolina, central part of Slovak Republic, *Lobaria pulmonaria* was collected in a mixed oak forest near Crevole, Murlo, Siena, Italy, *Stereocaulon tomentosum* was collected from the ground near the village of Gelnica-Cechy, *Xanthoria elegans* was collected during an expedition to Antarctica and *Cetraria islandica* was collected at Kojšovská hoľa near the village of Zlatá Idka from grassland, along with *Vaccinium vitis-ideaea* and *Vaccinium myrtillus*. All lichens were collected and identified by Dr. Dudáš and Dr. Goga and stored in a fridge for further experiments.

### 3.2. Lichen Extracts Preparation and Secondary Metabolites Isolation and Characterisation

Each lichen thalli (5 g/DW) were extracted in acetone in a glass beaker on a magnetic stirrer with a stirring bar for 30 min. The extracts were filtered and concentrated under reduced pressure in a rotary evaporator. This procedure was repeated twice. Dry extracts were stored in the refrigerator at 4 °C before screening assays. Lichen compounds were used from the internal bank of lichen secondary metabolites at the Department of Botany (Pavol Jozef Šafárik University in Košice, Faculty of Science). All secondary metabolites tested in these studies were identified and confirmed by HPLC and NMR, as published [29,47]. Overview of Lichen extracts and their secondary metabolites along with identification methods is given in Table 1. The stock solutions were prepared by dilution of dry powder in DMSO.

### 3.3. Cell Culture

The human cancer cell lines were purchased from ATCC (American Type Culture Collection; Manassas, VA, USA) or ECACC (European Collection of Authenticated Cell Cultures, Salisbury, UK). The HeLa (93021013, human cervical adenocarcinoma), HCT116 (CCL-247TM, human colorectal carcinoma), A2780 (93112519, Human ovarian carcinoma), Jurkat (88042803, human leukemic T cell lymphoma) were cultured in RPMI 1640 medium (Biosera, Kansas City, MO, USA) and A549 (CCL-185TM, human alveolar adenocarcinoma), MCF-7 (HTB-22TM, human Caucasian breast adenocarcinoma, ER+), Caco-2 (HTB-37TM, human colorectal adenocarcinoma), MDA-MB-231(HTB-26TM, human breast adenocarcinoma, ER-), BLM (human metastatic melanoma, a gift from prof. K. Smetana, Institute of Anatomy, Charles University in Prague) and COS-7 (CRL-1651TM, kidney fibroblasts) were maintained in growth medium consisting of high glucose Dulbecco’s Modified Eagle Medium (DMEM) + sodium pyruvate (Biosera). BJ-5ta (CRL-4001TM, human dermal fibroblasts) were maintained in growth medium consisting of high glucose DMEM:M199 4:1 medium mixture (Biosera) and Hygromycin B (0.01 mg/mL; Merck, Darmstadt, Germany). MCF-10A (CRL-10317TM, human mammary gland epithelial cells) were maintained in growth medium consisting of high glucose DMEM F12 Medium (Biosera) + Supplement (insulin, EGF- epithelial growth factor, HC-hydrocortisone, all Merck). All media were supplemented with a 10% fetal bovine serum (FBS; Gibco, Thermo Scientific, Rockford, IL, USA), Antibiotic/Antimycotic Solution (Merck) and maintained in an atmosphere containing 5% CO_2_ in humidified air at 37 °C. Cells were checked for mycoplasma using Hoechst 33342 staining (Merck).

### 3.4. 2D Culture and 3D Spheroids Preparation

For 2D adherent and suspension (Jurkat) cell culture, the cells were seeded into 96-well flat bottom microplates (TPP, Trasadingen, Switzerland) in concentrations adjusted to 5000–10,000 cells per well and maintained for 24 h until treatment. For spheroids (3D) formation, the cells (10,000/well) were seeded into 96-well round (U) bottom Nuclon^TM^ Sphera^TM^ microplates (Thermo Scientific, Rockford, IL, USA) with non-adhesive surface, enabling cell growth in suspension with virtually no cell attachment. The 3D conformation was visible 24 h after seeding and spheroids were maintained for 4 days until treatment with lichen extracts or secondary metabolites. Suspension Jurkat cells were also aggregated for 24 h, without disintegration after 4 days.

### 3.5. Resazurin Screening Assay

The anti-proliferative/cytotoxic effect of lichen extracts (final c = 10, 50, 100 µg/mL), secondary metabolites or cisplatine (c = 10, 50, 100 µM) and DMSO vehicle (*v*/*v* 0.02, 0.1, 0.2%) was determined by resazurin assay in HeLa, HCT116, Caco-2, MCF-7, MDA-MB-231, A549, BLM, A2780, Jurkat, COS-7, BJ-5ta and MCF-10A 2D and 3D cell cultures. After 24 h (2D) or 4 days (3D), final concentrations of lichen extracts and secondary metabolites prepared from DMSO stock solutions were added and incubation proceeded for the next 72 h in normal cultivation conditions (dark, 37 °C, 5% CO_2_). Ten or 15 microliters of resazurin solution (Merck, Darmstadt, Germany) at a final concentration of 40 µM was added to each well at the end-point (72 h). After a minimum of 1 h incubation, the fluorescence of the metabolic product resorufin was measured by the automated Cytation^TM^ 3 cell imaging multi-mode reader (Biotek, Winooski, VT, USA) with a 560 nm excitation/590 nm emission filter. The results were expressed as a fold of the control, where control fluorescence was taken as 100%. All experiments were performed in triplicate. The IC_50_ values were calculated from these data.

### 3.6. Antibacterial Activity of Lichen Extracts and Secondary Metabolites

#### 3.6.1. Microorganisms Used

The tested bacteria (Staphyloccocus aureus CCM 4223 and Escherichia coli CCM 3988) were obtained from the Czech collection of microorganisms (CCM, Brno, Czech Republic).

#### 3.6.2. Agar Well-Diffusion Method

The antibacterial properties of the seven extracts and four secondary metabolites *Lobaria pulmonaria*, *Cetraria islandica*, *Evernia prunastri*, *Stereocaulon tomentosum*, *Xanthoria elegans*, *Umbilicaria hirsute*, *Pseudevernia furfuracea*, gyrophoric acid, physodic acid, evernic acid and atranorin were evaluated by the agar well diffusion method with a slightly modified process compared to [52]. Firstly, each compound was dissolved in a small amount of 100% DMSO and then dissolved in 33.6 µM solution. As a positive control, gentamicin sulfate (Biosera, Nuaille, France) with a concentration of 50 µg/mL was used.

Bacteria were cultured overnight aerobically at 37 °C in LB medium (Sigma-Aldrich, Saint-Louis, MO, USA) with agitation. The inoculum from these overnight cultures was prepared by adjusting the density of culture to equal that of the 0.5 McFarland standard (1–2 × 10^8^ CFU/mL) by adding a sterile saline solution. These bacterial suspensions were diluted 1:300 in liquid plate count agar (HIMEDIA, Mumbai, India) resulting in a final concentration of bacteria of approximately 5 × 10^5^ CFU/mL and 20 mL of this inoculated agar was poured into a Petri dish (diameter 90 mm). Once the agar was solidified, five mm diameter wells were punched in the agar and filled with 50 μL of samples. Gentamicin sulphate (Biosera, Nuaille, France) with a concentration of 50 μg/mL was used as a positive control. The plates were incubated for 18–20 h at 37 °C. Afterward, the plates were photographed, and the inhibition zones were measured by ImageJ 1.53e software (U.S. National Institutes of Health, Bethesda, MD, USA). The values used for the calculation are mean values calculated from 3 replicate tests.

The antibacterial activity was calculated by applying the formula reported in (Rojas):%RIZD = [(IZD sample − IZD negative control)/IZD gentamicin] × 100(1)
where RIZD is the relative inhibition zone diameter (%) and IZD is the inhibition zone diameter (mm). As a negative control, the inhibition zones of 5% DMSO equal to 0 were taken. The inhibition zone diameter (IZD) was obtained by measuring the diameter of the transparent zone.

### 3.7. DPPH Radical Scavenging Assay

The free radical scavenging activity of lichen extracts was measured using 1,1-diphenyl-2-picryl-hydrazil (DPPH). The antioxidant activity via this method is described in several studies [53,54,55] and here it is slightly modified. The extracts were dissolved in a small amount of pure DMSO and then filled with distilled water to a concentration of 1 mg/mL. In addition, 10 μL of 5% NaOH solution was added to help dissolve the extracts fully. Stock solutions were subsequently diluted to the desired concentrations. The reaction mixture consisted of 2 mL of DPPH methanolic solution (0.1 mM) and 1 mL of our extracts to reach a final volume of 3 mL. The samples were incubated at laboratory temperature in the dark for 30 min. After the incubation, the absorbance of the samples was measured at 517 nm in a spectrophotometer (multi-detection microplate reader; Synergy HT, BioTek, Winooski, VT, USA). Ascorbic acid was used as positive control. A solution of 5% DMSO (1 mL) in MeOH (2 mL) was used as a blank control. The DPPH radical concentration was calculated by the equation:Free radical scavenging activity (%) = [(A0 − A1)/A)] × 100(2)
where A0 was the absorbance of negative control and A1 was the absorbance of the reaction mixture of our samples. The inhibition concentration at 50% inhibition (IC_50_) was the parameter used to compare the radical scavenging activity.

### 3.8. Statistical Analyses

Results are expressed as mean ± standard deviation (SD). Statistical analyses of the data were performed using standard procedures *t*-test. Differences were considered significant when *p* < 0.05. Throughout this paper * indicates *p* < 0.05, ** *p* < 0.01 versus untreated cells.

## 4. Conclusions

In summary, for the first time, we demonstrated the antiproliferative potential of some lichen extracts and their abundant secondary metabolites in robust 2D and 3D spheroid models. The result showed that the most potent lichen extracts were *Lobaria pulmonaria*, *Evernia prunastri*, *Stereocaulon tomentosum* and *Pseudevernia furfuracea* in both in vitro screening models. The differences between models in mostly occurred in secondary metabolites, where the most cytotoxic were physodic acid (2D model) and gyrophoric acid (3D model). Moreover, the 3D model compared to the 2D model showed increased cytotoxicity after DMSO or lichen extract/metabolites treatment, suggesting that morphology, physiology and distribution of substances/nutrients in the spheroid model compared to the cell monolayer are important factors influencing cytotoxicity results.

Furthermore, the antibacterial activity of lichen extracts as well as their secondary metabolites seem to be most effective against gram positive bacteria. Of all tested extracts, only *Xanthoria elegans* and *Umbilicaria hirsuta* did not show any antibacterial activity. Secondary metabolites also, except for physodic acid, did not show antibacterial potential. In the present study, all lichen extracts possessed antioxidant activity. The most promising scavenging activity was observed in the lichen *Lobaria pulmonaria*. It was also observed that *Xanthoria elegans* contains high amounts of pigments which may interfere with measurements of absorbance in DPPH assay. In further studies, a solution to this issue is worth considering.

For future in vivo experiments, complex screening assets, including the 3D spheroid model with analyses of antiproliferative, cytotoxic and redox status mediated properties should be taken to consideration.

## Figures and Tables

**Figure 1 plants-12-00611-f001:**
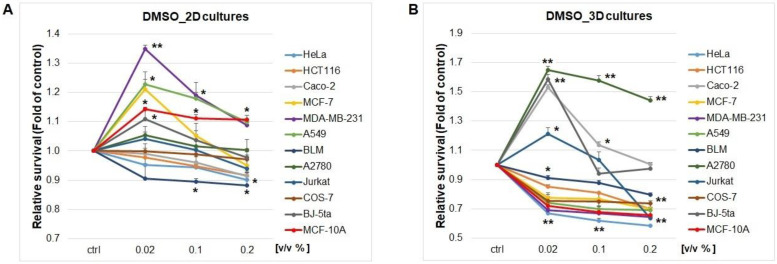
Metabolic activity influence (72 h) by DMSO vehicle in 2D (**A**) and 3D spheroid (**B**) cultures of various cancer and non-cancerous cell lines. Significance: * *p* < 0.05, ** *p* < 0.01 vs. untreated cells.

**Table 1 plants-12-00611-t001:** Overview of Lichen extracts and their secondary metabolites with identification method.

Lichens	Metabolites	Identification	References
*Pseudevernia furfuracea*	physodalic acid, atranorin, chloratranorin, 3-hydroxyphysodic acid, physodic acid	HPLC-UV, TLC, NMR	[19,20]
*Lobaria* *pulmonaria*	stictic acid, isidiophorin, rhizon-aldehyde, rhizonyl alcohol, pulmonarianin, vesuvianic acid, ergosterol peroxide	NMR, IR, UV, MS	[21]
*Cetraria* *islandica*	Succinprotocetraric acid, fumarprotocetraric acid, virensic acid, nephrosterinic acid, (+)- roccellaric acid, (+)-proto-lichesterinic acid, (+)-lichesterinic acid	UPLC-QTOF-MS	[22]
*Evernia prunastri*	salazinic acid, lecanoric acid, tetrahydroxy-tricosanoic acid, evernic acid, physodic acid, usnic acid, atranorin, chloratranorin, dihydrovinapraesoredisoic acid,	UPLC ESI-TOF-MS/MS	[23]
*Stereocaulon tomentosum*	atranorin, stictic acid, norstictic acid, constictic acid, consalazinic acid	HPLC-UV, TLC, NMR	[24,25,26,27]
*Xanthoria* *elegans*	physcion, emodin, physcion-bisanthrone, teloschistin monoacetate and derivates, parietin	HPLC-DAD	[28]
*Umbilicaria hirsuta*	gyrophoric acid, lecanoric acid	HPLC-UV, NMR	[29]

**Table 2 plants-12-00611-t002:** IC_50_ values (µg/mL or µM) list from 2D screening analyses.

2D	Cancer Cell Lines									Non-Cancerous Cell Lines		
	HeLa (Cervix)	HCT116 (Colon)	Caco-2 (Colon)	MCF-7 (Breast ER^+^)	MDA-MB-231 (Breast ER^−^)	A549 (Lung)	BLM (Melanoma)	A2780 (Ovarian)	Jurkat (Leukemia)	COS-7 (Kidney Fibroblasts)	BJ-5ta (Skin Fibroblasts)	MCF-10A (Breast Epithelial)
LOB	84.4	74.1	49.0	211.2	36.0	225.3	72.3	37.2	78.4	111.4	121.1	37.5
CET	98.5	152.2	286.1	520.9	298.2	206.0	162.0	26.8	67.1	466.6	469.9	74.0
EVER	43.5	44.6	128.0	103.2	75.3	90.0	45.0	70.7	88.5	96.9	69.7	39.8
STEREO	44.0	65.1	106.2	146.5	64.0	91.1	74.8	48.0	46.4	85.7	116.2	68.4
XANT	206.0	736.0	158.2	453.7	702.5	350.6	640.6	298.0	477.0	123.9	908.4	222.5
UMBILI	232.5	421.5	130.9	452.1	115.1	136.2	147.2	106.2	239.6	159.9	208.0	112.5
PSE	37.5	31.9	48.4	33.1	45.5	48.6	39.1	32.9	44.3	56.9	62.7	67.7
GA	287.0	1028.7	415.0	267.1	139.5	289.2	175.3	119.7	268.7	117.9	163.1	131.5
Phy	38.4	48.7	54.5	70.1	89.2	97.1	162.1	37.7	63.7	76.0	79.3	75.6
EA	781.5	375.6	244.7	268.8	726.6	204.9	659.6	95.9	458.0	292.5	512.8	561.1
A	146.7	152.9	381.5	421.6	233.8	212.7	155.7	212.1	340.4	242.2	162.7	315.2
CisPt	35.4	7.4	30.7	29.7	7.1	23.5	29.5	9.3	6.3	24.6	37.9	25.9
DMSO	740.0	706.8	578.9	245.2	292.0	516.1	217.8	1918.0	482.2	1594.4	420.9	1342.4

LOB—*Lobaria pulmonaria*; CET—*Cetraria islandica*; EVER—*Evernia prunastri*; STEREO—*Stereocaulon tomentosum*; XANT—*Xanthoria elegans*; UMBILI—*Umbilicaria hirsuta*; PSE—*Pseudevernia furfuracea*; GA—gyrophoric acid; Phy—physodic acid; EA—evernic acid; A—atranorin.

**Table 3 plants-12-00611-t003:** IC_50_ values (µg/mL or µM) list from 3D spheroids screening analyses.

3D	Cancer Cell Lines									Non-Cancerous Cell Lines		
	HeLa (Cervix)	HCT116 (Colon)	Caco-2 (Colon)	MCF-7 (Breast ER^+^)	MDA-MB-231 (Breast ER^−^)	A549 (Lung)	BLM (Melanoma)	A2780 (Ovarian)	Jurkat (Leukemia)	COS-7 (Kidney Fibroblasts)	BJ-5ta (Skin Fibroblasts)	MCF-10A (Breast Epithelial)
LOB	8.3	24.1	136.1	24.2	21.5	119.8	76.8	97.1	150.8	184.8	532.7	14.2
CET	268.9	182.1	263.5	407.0	187.6	312.7	300.0	1253.3	201.4	372.4	1299.2	282.4
EVER	46.7	123.0	181.7	65.5	67.0	66.1	123.9	551.2	112.6	144.2	183.0	127.9
STEREO	44.5	127.9	205.6	54.3	55.8	76.5	137.5	552.0	173.0	138.7	222.8	107.6
XANT	123.9	35.2	353.1	225.8	69.9	171.7	223.6	274.5	136.0	386.8	648.1	83.2
UMBILI	16.9	11.4	143.3	65.9	9.3	48.1	46.4	120.4	48.1	157.1	126.1	31.1
PSE	19.5	66.7	97.8	41.8	39.9	49.7	129.0	120.6	77.3	75.3	251.7	158.7
GA	17.9	93.6	45.1	67.8	73.5	29.1	105.1	148.6	37.4	93.2	211.8	64.1
Phy	272.1	511.0	647.9	195.0	112.2	214.8	296.1	365.9	263.5	363.3	263.2	1329.5
EA	156.4	281.4	893.8	265.2	152.4	273.9	157.7	263.9	219.8	260.7	189.8	387.7
A	146.0	122.2	230.9	309.8	154.2	152.1	311.4	244.5	225.1	1272.3	148.0	417.4
CisPt	78.0	98.4	183.8	78.3	75.4	119.3	89.9	134.3	6.7	153.8	186.5	121.3
DMSO	181.8	215.3	289.2	301.0	372.5	371.5	321.7	452.7	170.6	1630.5	742.0	314.9

LOB—*Lobaria pulmonaria*; CET—*Cetraria islandica*; EVER—*Evernia prunastri*; STEREO—*Stereocaulon tomentosum*; XANT—*Xanthoria elegans*; UMBILI—*Umbilicaria hirsuta*; PSE—*Pseudevernia furfuracea*; GA—gyrophoric acid; Phy—physodic acid; EA—evernic acid; A—atranorin.

**Table 4 plants-12-00611-t004:** Antibacterial activity of tested extracts and secondary metabolites.

	LOB	CET	EVER	STEREO	XANT	UMBILI	PSE	GA	PHY	EA	A
% RIZD *E. coli*	N.A.	N.A.	N.A.	N.A.	N.A.	N.A.	N.A.	N.A.	N.A.	N.A.	N.A.
% RIZD *S. aureus*	105.41	92.44	62.86	81.51	N.A.	N.A.	89.97	N.A.	118.78	N.A.	N.A.

LOB—*Lobaria pulmonaria*; CET—*Cetraria islandica*; EVER—*Evernia prunastri*; STEREO—*Stereocaulon tomentosum*; XANT—*Xanthoria elegans*; UMBILI—*Umbilicaria hirsuta*; PSE—*Pseudevernia furfuracea*; GA—gyrophoric acid; Phy—physodic acid; EA—evernic acid; A—atranorin.

**Table 5 plants-12-00611-t005:** IC_50_ by DPPH radical scavenging assay of tested extracts.

Extracts	IC_50_ (µg/mL)
*Evernia prunastri*	290.3 ± 79.3
*Stereocaulon tomentosum*	127.8 ± 26.3
*Lobaria pulmonaria*	24.8 ± 13.7
*Cetraria islandica*	532.3 ± 60.3
*Umbilicaria hirsuta*	286.7 ± 20.8
*Xanthoria elegans*	N.A.
*Pseudevernia furfuracea*	135.8 ± 6.9
Ascorbic acid	7.7 ± 0.4

## Data Availability

The data presented in this study can be provided by the authors by reasonable request.

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
