# Peer review of "Screening Evaluation of Antiproliferative, Antimicrobial and Antioxidant Activity of Lichen Extracts and Secondary Metabolites In Vitro"

_plants, 2023, doi:10.3390/plants12030611_

Round 1

Reviewer 1 Report

The present manuscript entitled “Antiproliferative, antimicrobial and antioxidative potential of lichen secondary metabolites in 2D and 3D in vitro screening models” by Martin Kello and coworkers can be reconsidered after major revisions.

In this study the potential antiproliferative, antimicrobial and antioxidative effects of several extracts from lichens and their metabolites were investigated and interestingly the potential activity on different cell lines were investigated by 2D and 3D cell models 

My observations and suggestions on the manuscript are as follows:

Title: the title in not clear, antimicrobial and antioxidant activity are in 2 D and 3D in vitro screening models? Please, rephrase the title 

Abstract

-       In the abstract it is reported that antioxidant activity was performed by DPPH and FRAP unlike in the text where only DPPH is present

Introduction

-       The introduction needs improvement. It is focused on tumors, chemotherapy, and efficacy of the 2D and 3D cell culture but does not mention the literature on antibacterial and antioxidant activity of lichens. Moreover, Authors should clearly define the objective of the article at the end of the paragraph having presented the state of the art and literature regarding the activity of lichens and their metabolites.

-       Clearly explain the reason for choosing the metabolites tested. Have you determined the chemical composition of the acetone extracts tested? Are these metabolites the most abundant compounds in the extracts? Do you want to demonstrate the activity could be related to the presence of such metabolites? And also, their role in the biological activity of the extracts should be discussed in the results. 

Results

- The manuscript focuses on studying the antiproliferative activity of the extracts, and both 2D and 3D cell cultures were considered to determine this activity. From my point of view, relying only on the resazurin assay and stating that some extracts have potential anti-cancer activity is excessive (page 3, lane 107); it would be better to define these extracts and physodic acid as cytotoxic. In addition, normal cell lines, particularly MCF-10, sometimes showed low IC50 values that denotes cytotoxicity of the samples even against healthy cell lines. To define a compound as selective, the ratio of the IC50 of the healthy cell line to that of the cancer cell line (selectivity index) for the substance studied must be above a certain value. I think the selectivity index must be introduced in the results.

- DMSO must be used as a vehicle at a concentration below the toxic concentration, and this concentration must be determined for each cell type. In my view the significant metabolic inhibition of DMSO in 3D cell culture affected the IC50 values obtained in the cell lines treatments.

- In table 1 and table 2 what is the unit for DMSO? In materials and method were reported the percentages. 

- The antibacterial activity was not discussed. Are in literature studies on the antimicrobial activities of lichens and their metabolites?

Material and methods: 

-       3.2. Lichen extracts preparation and secondary metabolites isolation and characterization: The metabolites tested in the research should be reported in this section. 

-       3.4 2D culture and 3D spheroids preparation: A clarification: can T lymphocyte cells, such as Jurkat cell line, form spheroids?

Conclusions

-       A conclusion paragraph should be added

Minor suggestions: 

-       In all the text, all scientific names must be written in Italics. Also, only the first time you use a scientific name of a plant or microrganism you should write the name in full. Thereafter, the name of the genus must be bulleted. 

-       In agar well diffusion method is not necessary to give the explanation of the acronym or the abbreviations, it has already been done. 

Author Response

We want to thank reviewer for valuable comments and advices how improve our manuscript. For our response see please attachment file.

Best regards

Martin Kello

Reviewer 2 Report

The manuscript plants-2087290 presents antiproliferative, antimicrobial and antioxidative potential of lichen secondary metabolites in 2D and 3D in vitro screening models. The article could be publised in Plants journal after a major revision.

As comments/sugestions:

1.How were the active compounds isolated from the structure of the tested lichens and how were the extracts administered for testing the biological activity obtained? 

2. What are the structures of the active principles extracted from the tested lichens?

3 What could be a possible mechanism of action (antitumor, antioxidant) for the active principles of the tested compounds?

Author Response

(The authors gave the same response as above.)

Reviewer 3 Report

This manuscript describes a complex investigation that provides new data on the potential antiproliferative, antimicrobial and antioxidant effects of several crude lichen extracts. It also provides insight into the secondary metabolites of lichens.

The issues addressed are relevant in the field and aim to fill some of the existing gaps given that the lichen compounds were neglected by the scientific community for long periods and nowadays are objects of investigation based on their promising effects.

The paper is well introduced and the methodology is well addressed and provided in sufficient detail. The references are relevant for this research topic, being written in agreement with the journal requirements.

The manuscript is very interesting, however there are some aspects that could improve it significantly:

·       I am aware of the word limit in an abstract, but I think it is important to add one or two relevant sentences at the end to make it more consistent.

·       At the beginning of section 2. Results and Discussion, lines 72-85 are more like a literature review. Please move this part to the Introduction section.

·       In the Results and discussion section, please highlight the antiproliferative effect as a separate part (for example 2.1. Antiproliferative activity).

·       Please expand the Antibacterial Activity and Antioxidant Activity sections and enhance them with more results-based discussions.

·       Sections 2.1 and 2.2 will become 2.2 and 2.3. I recommend to make possible correlations between antioxidant activity and antimicrobial and antiproliferative potential.

·       The conclusions section is missing. Please provide the conclusions of this paper in line with the evidence and arguments presented so that they answer the main question posed.

Author Response

(The authors gave the same response as above.)

Round 2

Reviewer 1 Report

Dear Authors, 

thank you for taking the suggestions.

Kind regards

Author Response

Dear reviewer,

thank you very much for the opportunity to improve our manuscript.

Best regards

Reviewer 3 Report

All comments and recommendations have been well addressed by the authors. The paper has been improved accordingly and can therefore be accepted for publication.

Author Response

(The authors gave the same response as above.)
